# Characterization of Ocular Surface Microbial Profiles Revealed Discrepancies between Conjunctival and Corneal Microbiota

**DOI:** 10.3390/pathogens10040405

**Published:** 2021-03-30

**Authors:** Anna Matysiak, Michal Kabza, Justyna A. Karolak, Marcelina M. Jaworska, Malgorzata Rydzanicz, Rafal Ploski, Jacek P. Szaflik, Marzena Gajecka

**Affiliations:** 1Chair and Department of Genetics and Pharmaceutical Microbiology, Poznan University of Medical Sciences, 60-781 Poznan, Poland; anna.matysiak@skpp.edu.pl (A.M.); mkabza@outlook.com (M.K.); jkarolak@ump.edu.pl (J.A.K.); marcelinajaworska@ump.edu.pl (M.M.J.); 2Institute of Human Genetics, Polish Academy of Sciences, 60-479 Poznan, Poland; 3Department of Medical Genetics, Medical University of Warsaw, 02-106 Warsaw, Poland; malgorzata.rydzanicz@wum.edu.pl (M.R.); rafal.ploski@wum.edu.pl (R.P.); 4Department of Ophthalmology, Medical University of Warsaw, 00-576 Warsaw, Poland; okovisus@gmail.com

**Keywords:** cornea, eye microbiome, corneal microbiota, conjunctival microbiota, RNA-Seq, host–pathogen interactions

## Abstract

The ocular microbiome composition has only been partially characterized. Here, we used RNA-sequencing (RNA-Seq) data to assess microbial diversity in human corneal tissue. Additionally, conjunctival swab samples were examined to characterize ocular surface microbiota. Short RNA-Seq reads, obtained from a previous transcriptome study of 50 corneal tissues, were mapped to the human reference genome GRCh38 to remove sequences of human origin. The unmapped reads were then used for taxonomic classification by comparing them with known bacterial, archaeal, and viral sequences from public databases. The components of microbial communities were identified and characterized using both conventional microbiology and polymerase chain reaction (PCR) techniques in 36 conjunctival swabs. The majority of ocular samples examined by conventional and molecular techniques showed very similar microbial taxonomic profiles, with most of the microorganisms being classified into *Proteobacteria*, *Firmicutes*, and *Actinobacteria* phyla. Only 50% of conjunctival samples exhibited bacterial growth. The PCR detection provided a broader overview of positive results for conjunctival materials. The RNA-Seq assessment revealed significant variability of the corneal microbial communities, including fastidious bacteria and viruses. The use of the combined techniques allowed for a comprehensive characterization of the eye microbiome’s elements, especially in aspects of microbiota diversity.

## 1. Introduction

Microbial communities represent an essential element of the human body. Consequently, some human metabolic features may be dependent on microbial traits. Thus, characterization of the microbiomes across the human body could be the first step in understanding the role of microorganisms in both health and disease [1]. Next-generation sequencing (NGS) technologies offer high-throughput analysis and the possibility of surveying fastidious and unculturable microorganisms [2]. The gut microbiome is the most known of all microbial communities studied in humans so far. [3,4]. In contrast, the human eye microbiome, including the composition of conjunctival and corneal microbiota, has been only partially investigated and insufficiently characterized [5].

A combination of microbial culture and DNA-sequencing techniques revealed that the ocular surface microbiota is rather paucibacterial [6]. Even when bacterial colonies are present, the number of colony-forming units (CFU) per conjunctival swab is usually much less than 100 CFU [7]. On the other hand, a study of the healthy ocular surface performed by sequencing of 16S rRNA amplicons generated from DNA extracted from conjunctival swabs revealed a diverse microbial community, including five phyla and 59 distinct bacterial genera [8]. Culture-independent microbial analyses revealed that more than 87% of conjunctival 16S ribosomal RNA (rRNA) sequences were mapped to three phyla: *Proteobacteria* (64.0–64.4%), *Actinobacteria* (15.0–19.6%), and *Firmicutes* (3.9–15.5%) [8,9]. Another study revealed that the most likely core of ocular surface microbiota contains commensal, environmental, and opportunistic pathogenic bacteria: *Pseudomonas*, *Corynebacterium*, *Acinetobacter*, *Staphylococcus*, *Streptococcus*, *Millisia*, *Anaerococcus*, *Finegoldia*, *Simonsiella*, and *Veillonella* [10]. To the best of our knowledge, there is no information concerning the corneal flora assessed in corneas incised during the penetrating keratoplasty procedure.

As the published investigations [11,12,13,14,15,16] focus rather on conjunctival, not corneal samples (Appendix A
Appendix A), the purpose of this study was to characterize and compare microbiota occurrence in the conjunctival and corneal samples. Here, bacterial, fungal, viral, and archaeal elements of the microbiome were investigated using various culture-based and molecular techniques.

## 2. Results

### 2.1. Characteristics of Bacterial, Archaeal and Viral Elements of the Human Corneal Microbiome Based on RNA-Sequencing Data

The most abundant microbial phylum detected in human corneal tissues was *Proteobacteria* (Figure 1). 

Within this phylum, *Alcanivorax* (12–22% of reads, 100% of examined samples) and *Pseudoalteromonas* (5–8% of reads, 100% of examined samples) were the most prevalent genera. Seven out of 50 corneal samples (KC_23, KC_26, KC_35, KC_37, KC_39, KR_26, KR_64) showed a distinct taxonomic profile characterized by a much higher percentage of reads belonging to the *Firmicutes* phylum (36–80% of reads). *Actinobacteria* were also present in corneal tissues, including *Nocardia* (detected in 29 samples) and *Actinomyces* (detected in one sample). The fraction of reads originating from other bacterial and archaeal phyla in corneal tissues was lower than 5% in the vast majority of samples and included *Cyanobacteria*, *Bacteroidetes*, *Fusobacteria*, and *Euryarchaeota*.

Reads of viral origin, mainly from Poxviridae and Herpesviridae families, contributed to 1–16% of all analyzed sequencing reads. Among them, 15–30% were classified above the phylum rank. Results of the overall RNA-Seq analysis are presented in Figure 2.

### 2.2. Characteristics of the Conjunctival Sample Microbiota

Microbial growth was observed in 75% of the samples tested by traditional cultivation techniques. The most frequently observed bacteria in the conjunctival swabs belonged to the *Firmicutes* phylum (Figure 3). *Staphylococcus epidermidis*, *Staphylococcus warneri*, *Staphylococcus haemolyticus*, and *Escherichia coli* were the most predominant species isolated from the ocular surface and were detected in 28%, 14%, 14%, and 11% of samples, respectively. Other Firmicutes members were present in less than 10% of samples and included *Staphylococcus aureus*, *Staphylococcus pasteuri*, *Staphylococcus capitis*, and *Kocuria* species (Figure 3). No growth of anaerobic bacteria or fungi was observed. The average number of microorganisms obtained from one swab did not exceed 20 CFU. In total, nine different species were detected among the 43 bacterial isolates from 36 ocular surface samples. Six out of 43 strains could not be identified at the genus level by the methods used.

*Staphylococcus* spp. was also ubiquitous among the examined conjunctival samples in the polymerase chain reaction (PCR) analyses. Other bacteria identified on ocular surface were *Moraxella catarrhalis*, *Ureaplasma* spp., *Propionibacterium* spp., and *Micrococcus* spp. (Figure 4). *Candida* spp., *Chlamydia* spp., *Mycoplasma* spp., *Streptococcus* spp., *Acinetobacter* spp., and *Pseudomonas* spp. were not detected using PCR.

### 2.3. Similarities and Discrepancies in Conjunctival and Corneal Microbial Profiles

In this study, with only a few exceptions, the analyzed samples showed very similar microbial taxonomic profiles, with most of the results being classified into three phyla: *Proteobacteria* (RNA-Seq: 30–63% of reads and 43% of positive results, conventional microbiology: 8% of positive results, PCR assays: 26% of positive results), *Firmicutes* (RNA-Seq: 2–17% of reads and 19% of positive results, conventional microbiology: 56% of positive results, PCR assays: 59% of positive results), and *Actinobacteria* (RNA-Seq: 1–17% of reads and 16% of positive results, conventional microbiology: 29% of positive results, PCR assays: 5% of positive results) (Figure 5). The PCRs and RNA-Seq analysis provided an overview of positive results for all conjunctival and corneal samples. The inherent human skin commensals were found in both conjunctiva (*Staphylococcus epidermidis*) and cornea (*Alcanivorax* spp.). No fungi were identified in conjunctival surface microbiota.

*Proteobacteria* was a dominant phylum in most of the corneal samples, while another phylum, *Firmicutes*, was detected in the vast majority of conjunctival samples. Only two genera, *Paenibacillus* (25–63%) and *Lactobacillus* (10–14%), belonging to *Firmicutes*, seem to be dominant in the microbiome of the seven corneal samples. However, these two were not detected in conjunctival swabs. While *Staphylococcus* spp. was predominant in conjunctival samples (56% in PCR), it constituted only 0.1% of reads (RNA-Seq) in corneal tissues. *Tenericutes* was a bacterial taxon only detected in the conjunctiva (10% of examined samples in PCR). In corneal tissues, we found fastidious microorganisms such as *Actinomycetales*, anaerobic bacteria, and viruses.

The results of questionnaire data analyses from corneas and conjunctivas swabs donors are presented in Appendix A.

## 3. Discussion

The microbial composition of the human eye plays an important role in both health and disease [17,18]. While there is a growing number of studies on the eye microbiome, questions about the microbial profile of the eye without symptoms of infection and its potential variation remain unanswered. In previously published reports, selected parts of the human eye and ocular area, i.e., external skin of the lid, the lid junction, conjunctival tissue [19], periocular skin, eyelid margin, or conjunctival fornix [20], were examined. Here, microbiome elements of the inner (cornea) and outer (conjunctiva) ocular surface were preliminarily characterized and compared to each other, according to phylum- and genera-level classification. 

In this study, the taxonomic composition of corneal and conjunctival microbiomes on the phylum level included *Proteobacteria*, *Actinobacteria*, and *Firmicutes,* which is consistent with the previously reported microbial profiles of the ocular surface [2,6,8,9,20]. On the other hand, we detected *Bacteroidetes* only in a small number of reads (4%) in our RNA-Seq study, while they were previously listed as a common bacterial taxon in ocular surface samples [5]. 

Consistent with reports, coagulase-negative *Staphylococci* (20–80%), including *Staphylococcus epidermidis*, were also detected in conjunctival samples of studied individuals [21,22]. Conjunctival swabs are technically challenging to obtain because of the contamination risk by microflora occupied nearest niches. This is why we could not exclude that identification of *Staphylococcus epidermidis*, a known inherent skin commensal, [23] could be a result of its translocation from skin to the eye during eye rubbing. This process could lead to the direct translocation of microorganisms from skin to the outer ocular surface and cause microdamage in the mechanical barrier, allowing microorganisms to enter the internal parts of the eye. Interestingly, *Staphylococcus epidermidis* was recognized in the conjunctival swabs derived from 17% of the individuals declaring a habit of eye rubbing, while *Alcanivorax* and *Pseudoalteromonas* were detected in corneas obtained from 20% of examined patients that reported eye rubbing (Appendix A
Appendix A). *Alcanivorax* and *Pseudoalteromonas* are known to be mostly marine, halophilic bacteria. However, they were also found in the salt-rich environment of the human skin [24,25]. Previously, it was hypothesized that the existence of skin-specific ecotypes of *Proteobacteria* might play a role in maintaining skin homeostasis and linking the environmental and human microbial communities [24]. While we could not exclude that the presence of *Alcanivorax* and *Pseudoaletromonas* in human corneas could be a result of skin-related sample contamination, the role of these bacteria should be further evaluated.

The small number of reads detected in corneal samples using RNA-Seq belongs to *Cyanobacteria*. *Cyanobacteria* were previously reported as an element of the human gut microbiome [26]. These bacteria were also detected in human skin as an effect of using skincare products containing plant material. Thus, indeed, the presence of *Cyanobacteria* in the eye specimens could be related to the skin–ocular surface contamination [27]. We have also experienced that the handling of samples with low microbial biomass results in data that are difficult to interpret, as previously indicated [28].

Prior studies demonstrated that the healthy ocular surface is paucibacterial. The lack of microbial diversity on the ocular surface could be explained by both innate antimicrobial factors and mechanical barriers (lysozyme, antimicrobial peptides, tear film, and blinking), which may limit the number of microorganisms [9,29]. Aseptic materials or an antibiotic administered prophylactically before ophthalmic procedures may further reduce the ocular surface’s amount and variety of microbiota [30]. In our studies, we anticipated differences in microorganism diversity in the assessed specimens. The preoperative antibiotic prophylaxis applied before the penetrating keratoplasty procedure could affect Gram-positive bacteria of the ocular surface microflora. Here, RNA from all corneal layers was extracted for the RNA-Seq analysis, and the antibiotic influence on microbiota of the deeper layers than the most external corneal epithelium remains to be elucidated. Nevertheless, we consider that preoperative prophylaxis may explain the presence of sparse reads originating from *Staphylococci* in corneal RNA-Seq data. Furthermore, all identified predominant genera also possess the ability to form biofilms, and forming biofilms with diminished biodiversity compared to the natural state might explain why only a handful of bacterial taxa were observed during the analysis [31].

Significant differences were observed between the microbial genus and species level detected in corneal and conjunctival samples. The variation of microbial composition in the assessed samples could be caused by differences between protocols used for sample collection and processing. Sampling the ocular surface with a swab for receiving conjunctival material can be managed with light or greater pressure. Therefore, different microflora can be derived depending on the swabbing technique used. The composition of samples derived by light swabbing is characterized by an overrepresentation of microbiota captured from the superficial layer, including *Firmicutes* (mainly *Staphylococci*), *Rothia* spp., *Herbaspirillum* spp., *Rhizobium* spp., and *Leptotricha* spp. In contrast, a deep swabbing procedure results in the growth of *Proteobacteria*, indicating an association with the conjunctival epithelium [7,8]. Here, we used the light swabbing technique to obtain conjunctival samples. Thus, identification of *Staphylococcus* spp. in these samples is not unexpected and is consistent with previously published data. On the other hand, RNA-Seq analyses of corneas showed an abundance of reads belonging to *Proteobacteria*, confirming the earlier results obtained for a deeper part of the ocular surface [7,8]. As, in corneas studied using RNA-Seq, we also detected bacteria belonging to *Firmicutes*, a phylum characteristic for the outer ocular surface, we hypothesize that corneal samples could be a better experimental material to analyze the integrity of eye microbiota. In addition, the applied advanced sequencing technique was more adequate in the microbiota identification process. 

Comparing the RNA-Seq and PCR-based results, the conventional culture-based methods had the smallest ability to detect elements of ocular surface microbiota. However, molecular testing also has limitations [32], i.e., using the RNA-Seq approach instead of 16S rRNA sequencing might have biased the obtained results [33]. In RNA-Seq data analysis, it is important to exclude erroneous background appearance in microbiota datasets [34] as was done for *Cutibacterium* spp., which was recognized in our study. Targeted PCR-based analyses provide significantly fewer data compared to the assessment performed by RNA-Seq. Using PCRs, we evaluated the most frequently cultured microorganisms derived from the ocular surface. In contrast, RNA-Seq allowed gaining information about viruses and fastidious and anaerobic bacteria, which gives a more detailed overview of diversity of the ocular surface microbiome [6,35].

Nevertheless, variability in the study subjects’ age and gender, as well as the fact that eye specimens were collected only once per each examined individual, constitute limitations of this study and might have influenced the recognized ocular surface’s microbiome composition, as indicated previously [19,20,21,36]. While the experimental protocol could be modified in the aspect of repeating assessment of conjunctival swabs, it was not possible to change it in the penetrating keratoplasty procedure. Furthermore, the incomplete questionnaire sections precluded us from estimating the influence of different factors, including traveling, medicine intake, book reading, and working in front of a computer, on the ocular surface microbiome composition.

As a microbiome consists of various elements, there is a need to extend research to study the bacterial element, as well as viruses, fungi, small eukaryotes, and archaea, which coexist and collaboratively act as societies in the particular human body parts. Here, *Cytomegalovirus* was one of the *Herpesviridae* found in 24% of corneal samples. Previous virus-directed studies have shown that herpes simplex type 1 virus, multiple sclerosis-associated retrovirus, human endogenous retrovirus K, torque teno virus, and even hepatitis viruses type B and C could be detected in tears or conjunctiva of healthy individuals. Their presence might suggest an additional potential niche for viruses at the ocular surface [6,17]. Additionally, some findings indicate that fungi such as *Aspergillus*, *Rhizopus*, *Penicillium*, or *Candida albicans* could be detected in ocular materials [7,9]. In our examined samples, fungi were not found.

## 4. Materials and Methods

### 4.1. Study Individuals and Materials

The study protocol was approved by the Institutional Review Board at Poznan University of Medical Sciences (453/14 and 755/19). All individuals provided informed consent after the possible consequences of the study were explained, in accordance with the Declaration of Helsinki. Furthermore, a questionnaire for each participant concerning gender and age, general and ocular illnesses, time spending in front of the computer and reading books, sport activity, medicines, travel, smoking, ocular complaints and surgeries, and ophthalmological examination data were completed. 

Full-thickness corneas were previously derived during the penetrating keratoplasty procedure performed in 50 nonrelated Polish individuals, as described previously [37]. It was not possible to collect conjunctival swab samples from these patients before corneal transplant surgery, which is a limitation of the study. Topical pilocarpine and 0.3% gentamicin were administered to the ocular surface into the conjunctival sack of the enrolled individuals before surgery in accordance with the hospital procedure. RNA samples from corneal tissues were previously extracted and the corneal transcriptome profile was previously assessed [37]. 

A total of 36 conjunctival samples were obtained from a study group consisting of 18 non-related Polish individuals, which did not undergo corneal transplantation, recruited independently from the 50 corneal transplant patients. The conjunctival samples were taken by trained laboratory diagnostician at the Poznan University of Medical Sciences according to a protocol established with a collaborating ophthalmologist by sampling the ocular surface, including inferior conjunctival fornices of both eyes with two separate swabs. An active ocular infection, inflammation, trauma, and lubricating or antibiotic eye drops usage were the exclusion criteria for participation in the study.

Culture swabs were sealed in a tube containing 1000 µL of transport medium (Copan, eSwabTM 481CE, Brescia, Italy). This system ensures the survival of aerobic, anaerobic, or demanding bacteria for the future culturing process. In addition, it provides stability of the collected genetic materials for 72 h at room temperature to be used in molecular analyses.

### 4.2. Identification of Bacterial, Archaeal and Viral Phyla in Corneal Samples

To characterize corneal microbiota composition and occurrence of the potential microbial variation between corneas of different individuals, RNA-Seq data were analyzed. To detect microbial phyla, we used a bioinformatics pipeline similar to the previously described [38]. BBDuk2 tool (http://jgi.doe.gov/data-and-tools/bbtools) was used to trim adapters and poor-quality regions (mean Phred quality score < 5) from Illumina short reads, as well as to remove reads matching human rRNA sequences. Reads shorter than 50 bp were excluded from the analysis. STAR was used to align filtered reads to the human reference genome (GRCh38/GENCODE 25, Ensembl 87) in the two-pass mode [39]. Unmapped read pairs were extracted using Sambamba package [40] and converted to FASTQ files using BEDTools [41]. Taxonomic classification of short read pairs was performed using Kaiju [42] by comparing them to the protein sequences from databases: proGenomes for bacteria and archaea, and RefSeq for viruses [43,44]. Greedy run mode with five allowed mismatches was used for maximum search sensitivity. 

### 4.3. Assessment of Bacterial and Fungal Cultures Derived from Conjunctival Samples

Standard microbiological culture-based methods [45] were applied to identify bacterial strains and fungi in material derived from the ocular surface. Conjunctival swab samples were separately inoculated onto plates with tryptase-soy agar (BioMèrieux, Marcy l’Etoile, France), mannitol salt agar (OXOID, Thermo Scientific, Basingstoke, UK), MacConkey agar (OXOID, Thermo Scientific), *Pseudomonas* cetrimide agar (OXOID, Thermo Scientific), and Sabouraud agar (OXOID, Thermo Scientific) and cultured under aerobic conditions at 37 °C. At the same time, another Sabouraud agar plate was cultured at 25 °C. In addition, samples were plated on blood agar and chocolate agar (BioMèrieux) plates following by incubation in 5–10% CO_2_ at 37 °C. For anaerobic bacteriological analysis, Schaedler agar (BioMèrieux) plates were used in anaerobic conditions at 37 °C. Swabs were also inoculated in Brain Heart Infusion (BHI) broth (OXOID, Thermo Scientific) to enhance the growth of potential fastidious bacteria. Inoculated plates were incubated for 48 h or longer, up to 7 days (the latter, especially for anaerobic bacterial and fungal cultures). Microorganisms were further evaluated using the semiautomatic microbial identification system—VITEK^®^2 Compact (BioMèrieux). Samples were recognized as a microbial culture negative when no growth was observed after 7 days of incubation.

### 4.4. Bacterial and Fungal Microbiome Elements—Molecular Detection in Conjunctival Samples

To characterize conjunctival microbiota elements, total genomic DNA was extracted from conjunctival swab specimens placed in 400 µL of transport medium, using the homemade protocol with phosphate-buffered sodium chloride and a heating process step. Detailed protocols of DNA extractions are available upon request. Multiple PCRs were performed to identify *Staphylococcus* spp., *Streptococcus* spp., *Micrococcus* spp., *Escherichia* spp., *Acinetobacter* spp., *Pseudomonas* spp., *Mycoplasma* spp., *Ureaplasma* spp., *Chlamydia* spp., *Propionibacterium* spp., and *Candida* spp. genera. In addition, the species of *Staphylococcus aureus*, *Staphylococcus epidermidis*, *Staphylococcus pasteuri*, *Staphylococcus warneri*, *Staphylococcus capitis*, *Staphylococcus haemolyticus*, *Streptococcus pneumoniae*, *Moraxella catarrhalis*, *Escherichia coli*, *Propionibacterium acnes*, *Candida albicans*, and *Candida parapsilosis* were investigated. The applied PCR primers are listed in Appendix A. Negative controls were included in each PCR experiment to exclude reagent or environmental contamination.

## 5. Conclusions

Summarizing, similarities and discrepancies found between the examined eye specimens indicate a further need to characterize the elements of the ocular surface microbiome core. The data obtained here should be additionally assessed, especially in aspects of eye microbiota interactions in ocular diseases with idiopathic and complex etiology and in complications of intraocular surgeries. Furthermore, a future longitudinal study should be undertaken to distinguish microorganisms at the other eye parts. Our data indicated microbiota diversity in the eye specimens, which ought to be further investigated using the most effective identification methods in the representative group of subjects, simultaneously in various eye tissues and considering environmental factors influencing microbiota composition.

## Figures and Tables

**Figure 1 pathogens-10-00405-f001:**
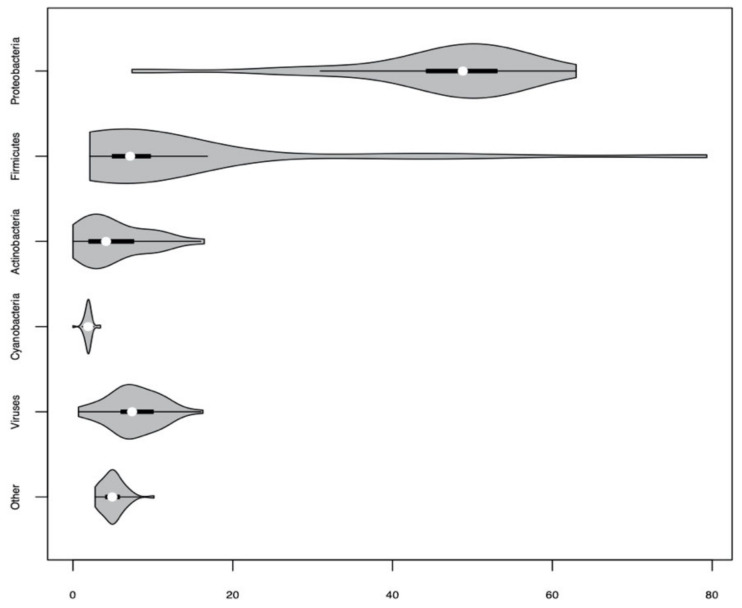
Violin plot (box plots with rotated kernel density plots added on each side) presenting the percentage of classified short reads assigned to different phyla in 50 human corneas based on the RNA-Seq data.

**Figure 2 pathogens-10-00405-f002:**
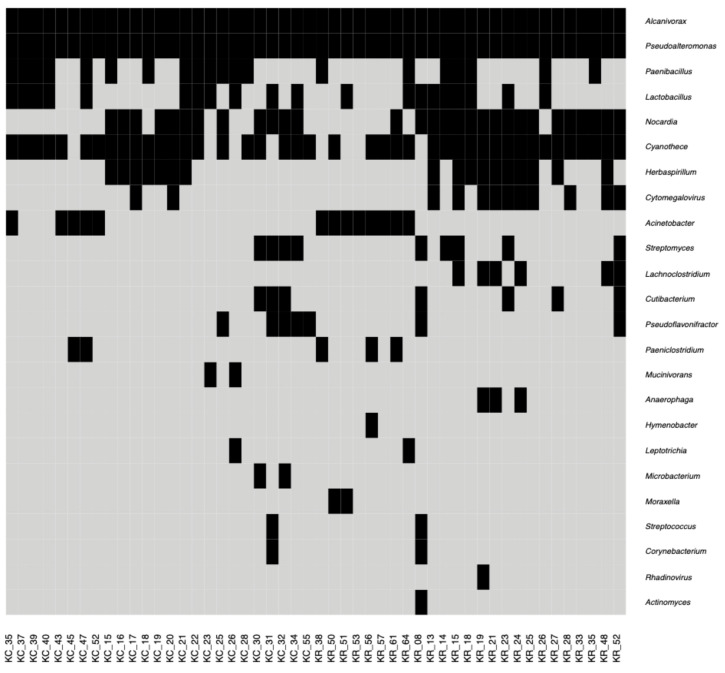
Heatmap showing presence (black) or absence (light gray) of different prokaryotic and viral genera in 50 human corneas based on the RNA-Seq data. Presence of a genus is defined by the threshold value of 1% of classified reads assigned to it.

**Figure 3 pathogens-10-00405-f003:**
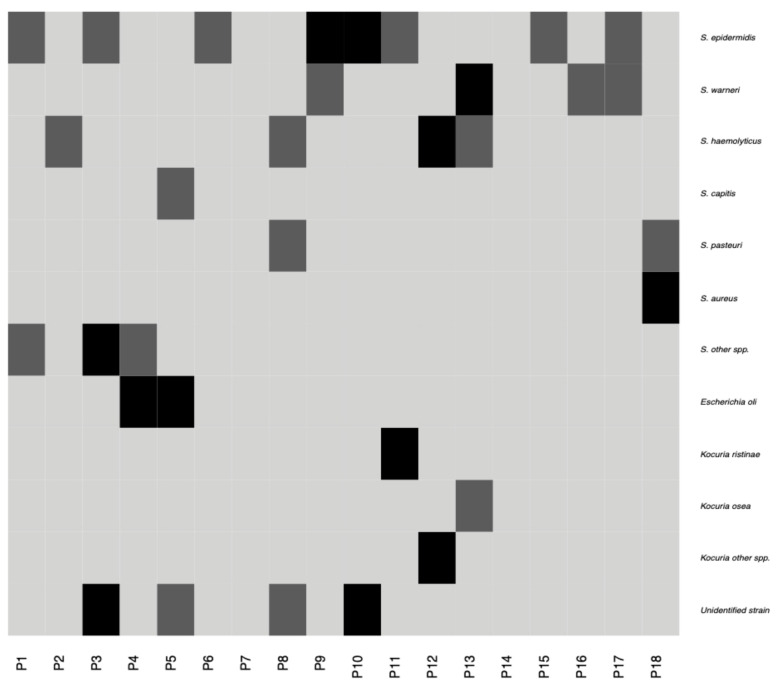
Heatmap showing presence and absence of different bacterial species/genera in 18 patients established using traditional microbiology techniques. Black and dark-gray colors are used to designate presence of a given taxon in both eyes and one eye respectively, while light-gray color indicates taxon absence. Negative fungal results are not presented.

**Figure 4 pathogens-10-00405-f004:**
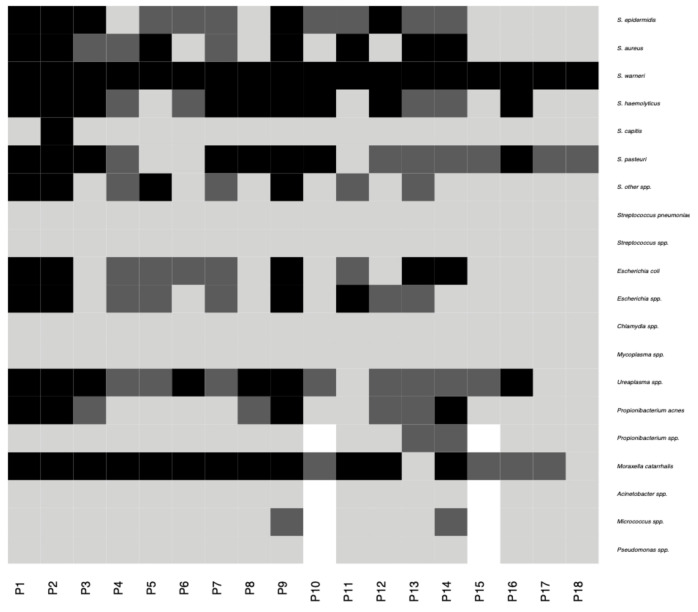
Heatmap showing presence and absence of different bacterial species/genera in 18 patients established using PCR methods. Black and dark-gray colors are used to designate presence of a given taxon in both eyes and one eye, respectively, while light-gray color indicates taxon absence. Missing values are colored white. Negative fungal results are not included.

**Figure 5 pathogens-10-00405-f005:**
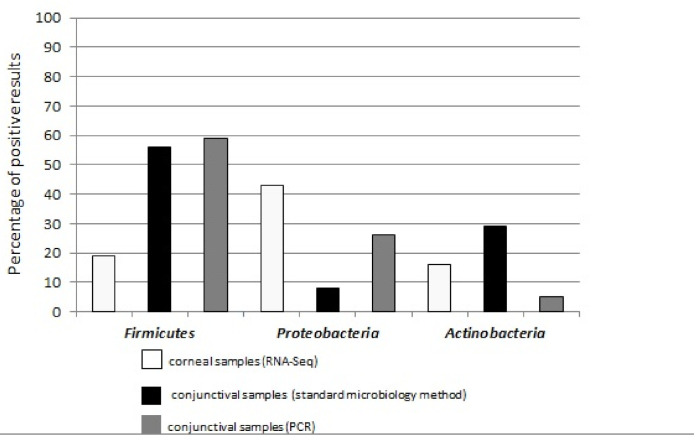
Microbial phyla profiles found using three detection methods (RNA-Seq, conventional microbiology, and PCR assays) with majority of microbiota being classified into three phyla: *Proteobacteria*, *Firmicutes* and *Actinobacteria*.

## Data Availability

RNA-sequencing data are available in the Gene Expression Omnibus database (GSE77938).

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
