# Peer review of "Characterization of Ocular Surface Microbial Profiles Revealed Discrepancies between Conjunctival and Corneal Microbiota"

_pathogens, 2021, doi:10.3390/pathogens10040405_

Round 1
Reviewer 1 Report
The authors of the present study examined samples taken from the conjunctivas of 50 patients and corneas of 36 patients for microbial composition. Interestingly, they found differences in the microbiota between the samples taken from the cornea versus the conjunctiva. It was never really clearly stated in what clinical environment the samples from the ocular surfaces were taken in, but it seems to be insinuated that they were taken from patients prior to penetrating keratoplasty. How were the patients identified for participation in the study? Were cornea and the conjunctival samples taken from the same patient or were the samples from each environment from separate patients? If there were from separate patients that could drastically influence whether similar or dissimilar microbial flora would be present from each portion of the ocular surface and if they were from separate patients, why? Also, it seems to be stated in the manuscript, but also unclear, that the samples were taken AFTER gentamicin was administered to the eyes? This would severely question the legitimacy of the results from the study and should be addressed by the authors. Were any other eye drops administered before the samples were taken from the eyes? If so, that could influence the results of the cultures and molecular techniques by cross contaminating surface microbes between the cornea and conjunctiva. Studies describing the microbiota of the ocular surfaces are important to better understand the beneficial and pathological role of microbes on ocular health and disease and risk factors for severe infections. The manuscript presents interesting information; however, there are several important questions as mentioned above that need to be answered before I could recommend the manuscript for publication as well as extensive revision due to grammar, punctuation and English errors. The results as well as the materials and methods of the study are difficult to understand at times due to the errors. The authors also mentioned variability in previous studies in the microbes found based on the technique of collecting the sample. Was a single individual in charge of collecting every sample to limit this variability or were multiple individuals trained to insure similar technique? It is difficult to recommend the manuscript for publication until the questions above are answered. I would really like a better description of the selection process of the subjects and of the methods of the manuscript.
Below are examples of grammatical errors in the manuscript, but is not a comprehensive list by any means.
Lines 40-42 – I do not understand this sentence. Needs grammar correction.
Line 79 – what is the purpose of the “ ‘ “ at the end of corneal
Lines 98-99 – this sentence is also confusing
Lines 100-101 – sentence needs revising
Figure 3 – I found it interesting that the majority of culture-positive samples demonstrated unilaterality in species found in each eye.
Line 107 – polymerase and analyses
Line 117 – I think the word “several” should probably be replaced by “few” since the word several generally implies a larger amount and the authors are implying that there was not much difference.
Line 158 – should be “consistent”
Lines 161 – 163 – Should be revised similarly to - “This is why we could not exclude that identification of Staphylococcus epidermidis, a known inherent skin commensal, [23] could be a result of its translocation from skin to eye during eye rubbing.”
Line 171 – salt-rich
Line 173 – delete “the” before maintaining.
Line 177 – the phrasing of this sentence is also a bit confusing.
Line 189 – surface’s
Lines 198-200 – both sentences need revision
Lines 223 – analysis
Line 230 – “might have”, “surface’s”
Lines 271-272 - analysis
Lines 190-192 – so the eye was treated with antimicrobials?
Reviewer 2 Report
The authors have evaluated the microbial profiles and cornea and conjunctiva. I have concerns about the study methodology and these need to be addressed prior to publication.
- Why did the authors not use RNA-Seq to profile conjunctival microbiome?
- Corneal samples seem to have been collected from diseased or abnormal corneas, it is quite probable that this can affect the microbial profile, yet no explanation or justification has been provided towards this fact.
- Why did the authors not use universal primers which could have provided a better characterization of the conjunctival microbiota?
- It is also quite surprising that marine bacteria were detected in the ocular samples while no fungal species were detected.
- Were any negative controls used during the sampling process?
Round 2
Reviewer 1 Report
I would like to thank the authors for addressing all of my concerns with their thorough response. The study is interesting and describes an important microbiome that is poorly studied to this date. Even though the conjunctiva and corneal samples were taken from separate patients and gentamicin was applied prior to sample collection, the information in the manuscript is still important and describes the microbiota of the ocular surface prior to corneal transplantation. The authors clearly state this fact in the limitations of their study. There are still some minor typos and grammatical errors throughout the paper, but the methods and data are adequately described. I would recommend the manuscript for publication if careful revision of the manuscript for grammatical errors was performed.
Author Response
We would like to thank Reviewer 1 for constructive and helpful comments. We revised our manuscript according to Reviewer comments. We highlighted typos and grammatical errors changes. No other changes have been made.
We modified the text in Abstract (Lines: 15-18, 26, 28-19), Introduction (Lines: 39-41, 46, 60-61), Results (Lines: 64, 82, 90-95, 106, 109-110, 117, 128-129), Discussion (Lines: 144-145, 154, 156, 163-166, 179-180, 191, 193-195, 205-206, 211, 214, 218, 225-226, 228-230, 231, 233-236, 238, 240-243), Materials and Methods (Lines: 256, 260, 262, 266, 272-273, 284, 306, 314-316) and Conclusions (Line 336).